# Effectiveness of Messenger RNA Vaccines against SARS-CoV-2 Infection in Hemodialysis Patients: A Case–Control Study

**DOI:** 10.3390/vaccines11010049

**Published:** 2022-12-26

**Authors:** Mohamad M. Alkadi, Abdullah Hamad, Hafedh Ghazouani, Mostafa Elshirbeny, Mohamed Y. Ali, Tarek Ghonimi, Rania Ibrahim, Essa Abuhelaiqa, Abdul Badi Abou-Samra, Hassan Al-Malki, Adeel A. Butt

**Affiliations:** 1Department of Medicine, Division of Nephrology, Hamad Medical Corporation, Doha P.O. Box 3050, Qatar; 2Department of Quality and Patient Safety, Hamad Medical Corporation, Doha P.O. Box 3050, Qatar; 3Departments of Medicine and Population Health Sciences, Weill Cornell Medicine, New York, NY 10065, USA; 4Departments of Medicine and Population Health Sciences, Weill Cornell Medicine, Doha P.O. Box 3050, Qatar

**Keywords:** COVID-19, dialysis, vaccine

## Abstract

Patients with end-stage kidney disease (ESKD) are at increased risk for SARS-CoV-2 infection and its complications compared with the general population. Several studies evaluated the effectiveness of COVID-19 vaccines in the dialysis population but showed mixed results. The aim of this study was to determine the effectiveness of COVID-19 mRNA vaccines against confirmed SARS-CoV-2 infection in hemodialysis (HD) patients in the State of Qatar. We included all adult ESKD patients on chronic HD who had at least one SARS-CoV-2 PCR test done after the introduction of the COVID-19 mRNA vaccines on 24 December 2020. Vaccinated patients who were only tested before receiving any dose of their COVID-19 vaccine or within 14 days after receiving the first vaccine dose were excluded from the study. We used a test-negative case–control design to determine the effectiveness of the COVID-19 vaccination. Sixty-eight patients had positive SARS-CoV-2 PCR tests (cases), while 714 patients had negative tests (controls). Ninety-one percent of patients received the COVID-19 mRNA vaccine. Compared with the controls, the cases were more likely to be older (62 ± 14 vs. 57 ± 15, *p* = 0.02), on dialysis for more than one year (84% vs. 72%, *p* = 0.03), unvaccinated (46% vs. 5%, *p* < 0.0001), and symptomatic (54% vs. 21%, *p* < 0.0001). The effectiveness of receiving two doses of COVID-19 mRNA vaccines against confirmed SARS-CoV-2 infection was 94.7% (95% CI: 89.9–97.2) in our HD population. The findings of this study support the importance of using the COVID-19 mRNA vaccine in chronic HD patients to prevent SARS-CoV-2 infection in such a high-risk population.

## 1. Introduction

Severe acute respiratory syndrome coronavirus 2 (SARS-CoV-2) is a single-stranded mRNA virus that was first identified in China and resulted in a worldwide pandemic that began in March 2020. Since then, hundreds of millions of people became infected with SARS-CoV-2, and more than 4 million have died due to coronavirus disease (COVID-19) [1]. The clinical presentation of SARS-CoV-2 infection has been widely variable, ranging from asymptomatic or mild symptoms (80%) to severe or life-threatening disease [2]. Several studies showed that dialysis patients were more vulnerable to SARS-CoV-2 infection and more likely to have a severe presentation, with mortality rates exceeding 20% [3,4]. Therefore, developing an effective vaccine in such a high-risk population has been essential and a priority for researchers and healthcare professionals working in the field. The earliest COVID-19 mRNA vaccines to obtain authorization were the BioNTech-162b2 (Pfizer) and the mRNA-1273 (Moderna) vaccines, with ≥95% efficacy in preventing illness and severe complications, including mortality, in the general population [5,6,7,8]. However, dialysis patients were excluded from those earlier studies for safety considerations.

The response of patients with advanced kidney disease to different vaccines, such as hepatitis B and influenza, and their ability to produce sufficient antibodies are more variable and less predictable than the general population [9,10]. One concern was whether their response to COVID-19 vaccines would be less than the general population. As COVID-19 vaccines became more widespread, data emerged about their efficacy in dialysis patients. Although initial studies showed heterogenous results regarding the vaccine efficacy in dialysis patients, subsequent studies were more promising and showed a good response [11,12,13]. Despite the effectiveness of COVID-19 vaccines in HD patients, several side effects were reported, such as fever, fatigue, myalgia, arthralgia, headache, syncope, and pericarditis [12]. The aim of the current study was to determine the effectiveness of COVID-19 mRNA vaccines against confirmed SARS-CoV-2 infection in chronic hemodialysis (HD) patients in the State of Qatar.

## 2. Materials and Methods

### 2.1. Study Population and Design

COVID-19 vaccination first became available in Qatar on 24 December 2020. The BioNTech-162b2 (Pfizer BioNTech) and the mRNA-1273 (Moderna) have been the only available vaccines in Qatar. All COVID-19 vaccines have been administered solely by the government since the start of the vaccination campaign. Immunocompromised patients, such as HD patients, transplant recipients, and cancer patients, were given the highest priority to get vaccinated. Until 5 January 2022, polymerase chain reaction (PCR) was the only accepted test to diagnose SARS-CoV-2 infection in Qatar. All information related to the vaccine, such as the type of vaccine (Pfizer or Moderna), the number of doses delivered and the date of administration, and all SARS-CoV-2 PCR results had to be recorded in a national-based electronic medical record system (Cerner-North Kansas City, MO, USA).

Hamad Medical Corporation is the sole provider of ambulatory dialysis services in the State of Qatar. We initially reviewed all SARS-CoV-2 polymerase chain reaction (PCR) tests done in our adult (≥18-year-old) chronic HD patients before 3 January 2022. Chronic HD was defined as being on HD for ≥3 months. We then stratified patients according to their vaccination status into vaccinated and unvaccinated. In this study, we included vaccinated patients who had a SARS-CoV-2 PCR test ≥14 days after the first dose of the vaccine and unvaccinated patients who had a PCR test done after the introduction of COVID-19 vaccination on 24 December 2020. Unvaccinated patients who only had a SARS-CoV-2 PCR test done before 24 December 2020 and vaccinated patients who had a test done before their first vaccine dose or within 14 days after the first dose were excluded from the study. We then categorized patients who met the inclusion criteria into cases and controls. Cases included both unvaccinated patients with a positive SARS-CoV-2 PCR test after 24 December 2020 and vaccinated patients with a positive test ≥14 days after their first vaccine dose. On the other hand, controls included both unvaccinated patients with a negative SARS-CoV-2 PCR test after 24 December 2020 and vaccinated patients with a negative test ≥14 days after their first vaccine dose. The effectiveness of the COVID-19 mRNA vaccine against confirmed SARS-CoV-2 infection was determined using a test-negative case–control design as it is a widely accepted standard to assess vaccine effectiveness in a population after introducing a vaccine [8,14,15,16,17]. The study design is summarized in Figure 1. 

Various clinical and laboratory parameters were collected from the national-based electronic medical record (Cerner-North Kansas City, MO, USA). These parameters included age, gender, race, duration of dialysis, comorbidities, type of COVID-19 vaccine given, number and dates of vaccine doses received, dates and results of SARS-CoV-2 PCR tests done, and presence or absence of symptoms at the time of PCR tests. This study was approved by the Institutional Review Board of Hamad Medical Corporation with a waiver of informed consent (MRC-05-161) given the study’s retrospective design.

### 2.2. Statistical Analysis

Data were summarized as a frequency with a percentage for categorical variables and a mean with a standard deviation for continuous variables. Chi-square or Fisher’s exact test was applied to categorical variables wherever appropriate, while an unpaired t-test was performed for continuous variables. A *p*-value of less than 0.05 was used for the statistically significant level. We used conditional logistic regression to calculate the odds of testing positive among the vaccinated versus unvaccinated HD patients. Vaccine effectiveness was determined using the following formula: [vaccine effectiveness = 1 – odds (vaccinated patients infected with SARS-CoV-2 |total number of vaccinated patients) /odds (non-vaccinated patients infected with SARS-CoV-2 | total number of non-vaccinated patients)].

## 3. Results

### 3.1. Baseline Characteristics 

Between 29 February 2020 and 3 January 2022, 6611 SARS-CoV-2 PCR tests were performed on 1329 HD patients. SARS-CoV-2 PCR tests were done for clinical suspicion in symptomatic patients, routine screening (prehospitalization, pre-procedures, before and after travel), or contact tracing of SARS-CoV-2-positive cases. Of the 1329 tested patients, 782 met the study’s inclusion criteria; 68 (9%) had positive SARS-CoV-2 PCR tests (cases), while 714 (91%) had negative SARS-CoV-2 PCR tests (controls) (Figure 1). Sixty-four percent of the study population were males, and 85% were older than 40 years. Most patients received HD for more than one year (73%) and had three or more comorbidities (88%). Hypertension was the most common comorbidity in patients (99%), followed by diabetes mellitus (66%). Only two patients were on corticosteroids, and both were in the cases group. Compared with the controls, the cases were more likely to be older (62 ± 14 vs. 57 ± 15, *p* = 0.02), on dialysis for more than one year (84% vs. 72%, *p* = 0.03), unvaccinated (46% vs. 5%, *p* < 0.0001), and symptomatic (54% vs. 21%, *p* < 0.0001). The baseline characteristics of cases and controls are summarized in Table 1.

### 3.2. Vaccine Effectiveness

Of the 782 HD patients included in the study, 715 received the COVID-19 mRNA vaccine (91%). Ninety-nine percent of the vaccinated patients had at least two doses of the COVID-19 mRNA vaccine (*n* = 707), and most patients received the Pfizer BioNTech (BNT162b2) vaccine (90%). The median time between the first and second doses of BNT162b2 was 21 days compared with 28 days for the Moderna (mRNA-1273) vaccine. The median time between the second dose of vaccine and SARS-CoV-2 infection was 282 days (IQR: 142-318). The overall vaccine effectiveness in patients that received two doses of the COVID-19 mRNA vaccine was 94.7% (95% CI 89.8-97.2). The vaccine effectiveness was similar for both the BNT162b2 (94.3%) and the mRNA-1273 vaccines (98.2%), as shown in Table 2.

Dialysis patients started receiving the third dose of the COVID-19 mRNA vaccine in August 2021; by 3 January 2022, 347 patients (49%) had received the third dose. Only 52 patients had SARS-CoV-2 PCR tests done after receiving their third dose: 9 had positive results, while 35 had negative results. The overall vaccine effectiveness in patients that received three doses of the COVID-19 mRNA vaccine was 70.1% (95% CI 23.3-89).

## 4. Discussion

Older age was reported as one of the nonresponse factors to COVID-19 vaccinations in previous studies of dialysis patients [13]. In our study, we found a higher rate of SARS-CoV-2 infection in older HD patients, which might have been due to a deficiency in both arms of immunity, namely, humoral and cell-mediated, in dialysis patients [10].

Most patients in our study were asymptomatic (76%), and they were tested for contact tracing or as a routine screening pre-travel, pre-procedure, or pre-hospitalization. Forty-six percent of patients with confirmed SARS-CoV-2 infection (cases) were asymptomatic. Our findings are similar to other studies that showed up to 50% of dialysis patients with SARS-CoV-2 infection were free of symptoms [18,19]. However, having symptoms was statistically significant in our cases compared with the controls (54% vs. 21%, *p* < 0.0001). Most symptoms ranged between fever and cough, but some patients developed breathing difficulties.

Seroconversion and antibody detection were used as surrogate markers of COVID-19 vaccine effectiveness in several clinical trials; however, there are still no widely accepted standards regarding the desired antibody titers or the assays used to measure SARS-CoV-2 antibodies [20]. In a recent meta-analysis of 33 studies designed to assess COVID-19 vaccine efficacy in dialysis and CKD patients, the development of antibodies was used in 22 studies. Most studies did not compare the antibody responses to appropriate controls, as 10 different assays had been used, making the interpretation of antibody levels difficult [13]. For this reason, we preferred to use confirmed SARS-CoV-2 infection as a marker of vaccine efficacy rather than seroconversion and antibody titers.

The efficacy of mRNA-1273 and BNT-162b2 vaccines against confirmed SARS-CoV-2 infection in our HD patients exceeded 90%, which supported the accumulated evidence of the effectiveness of COVID-19 vaccines in dialysis patients [11,21,22]. It also demonstrated the importance of using COVID-19 vaccines in such patients to decrease the risk of developing COVID-19.

The strength of this study was in having a unique registry system of vaccinations, infection confirmation, and hospital admission that guaranteed data accuracy since the start of the COVID-19 pandemic. The study design was also based on confirmed SARS-CoV-2 infection in assessing vaccine effectiveness rather than antibody seroconversion. To our knowledge, this is the first study in the Middle East region with a multiethnic and diverse population background. Our study had some limitations, such as having a small sample size and mainly testing symptomatic patients and those with recent contact with positive cases. Some asymptomatic patients could have been missed, resulting in an underestimation of the disease burden. None of the patients in our study had their antibody titers measured post-vaccination due to the lack of widely accepted standards regarding antibody titers. In addition to this, COVID-19 mRNA vaccines were the only available vaccines in Qatar. Therefore, no conclusion can be drawn regarding the superiority of the COVID-19 mRNA vaccine relative to other types of COVID-19 vaccines at resisting the development of COVID-19 in chronic HD patients.

## 5. Conclusions

COVID-19 mRNA vaccines were highly effective at resisting the development of COVID-19 in ESKD patients receiving chronic HD. Therefore, HD patients should be encouraged to receive COVID-19 vaccination since they are at higher risk for having severe complications due to COVID-19 compared with the general population. Widely using COVID-19 mRNA vaccines may help to reduce the risk of disease transmission within dialysis units, hospitalizations, and death among dialysis patients.

## Figures and Tables

**Figure 1 vaccines-11-00049-f001:**
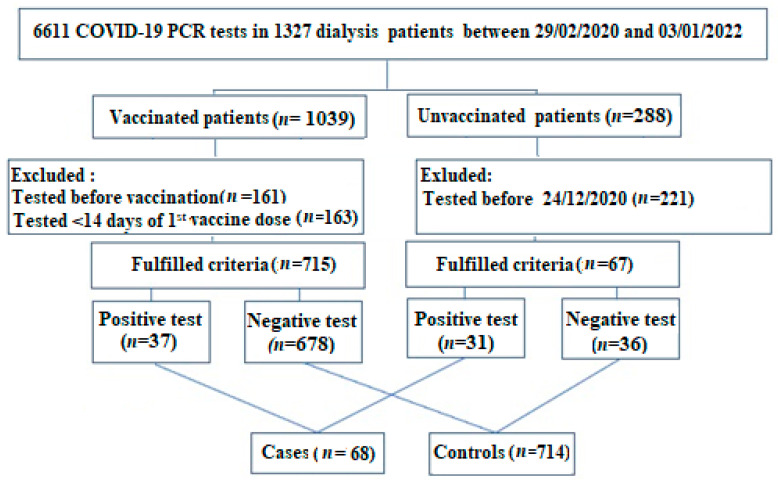
Flow chart of the study design.

**Table 1 vaccines-11-00049-t001:** Baseline characteristics of cases and controls.

Parameter	Total		Controls		Cases		*p*-Value
	(*n* = 782)		(*n* = 714)		(*n* = 68)		
	*n*	%	*n*	%	*n*	%	
**Age, years**							0.02
18–≤40	115	15	112	16	3	4	
41–<65	448	57	400	56	48	71	
≥65	219	28	202	28	17	25	
Mean ± SD	57 ± 15		57 ± 15		62 ± 14		
**Gender**							0.36
Male	501	64	454	64	47	69	
Female	281	36	260	36	21	31	
**Race**							0.22
Asian	681	87	625	88	56	82	
Non-Asian	101	13	89	12	12	18	
**Duration of dialysis**							0.03
<1 year	214	27	203	28	11	16	
≥1 year	568	73	511	72	57	84	
**Comorbidities**							
Hypertension	771	99	705	99	66	97	0.39
Diabetes	518	66	471	66	47	69	0.60
Cerebrovascular disease	72	9	62	9	10	15	0.10
Malignancy	35	4	29	4	6	9	0.07
COPD	36	5	33	5	3	4	0.94
Liver disease	39	5	34	5	5	7	0.36
**Number of comorbidities**							0.65
1–2	94	12	87	12	7	10	
≥3	688	88	627	88	61	90	
**Vaccination status**							<0.0001
Pfizer (BNT162b2)	640	82	607	85	33	49	
mRNA-1273 (Moderna)	75	9	71	10	4	6	
Not vaccinated	67	9	36	5	31	46	
**Symptomatic**							<0.0001
Yes	185	24	148	21	37	54	
No	597	76	566	79	31	46	

**Table 2 vaccines-11-00049-t002:** Vaccine effectiveness against infection in hemodialysis patients.

Vaccine Type	Cases	Controls	Adjusted	(95% CI)
	(*n* = 68)	(*n* = 714)	VE *, %	
**One Dose**				
Moderna	2	2	<0	(<0–92.1)
Pfizer	1	13	91.1	(32.7–99.8)
Overall	3	15	76.8	(5.5–96.0)
**<14 Days after the Second Dose**				
Moderna	1	1	<0	(<0–98.9)
Pfizer	3	1	<0	(<0–99.5)
Overall	4	2	<0	(<0–96.3)
**>14 Days after the Second Dose**				
Moderna	1	68	98.2	(88.6–99.9)
Pfizer	29	593	94.3	(89.0–97.0)
Overall	30	661	94.7	(89.9–97.2)
**Unvaccinated**	65	223	Reference	Reference

* Estimated as (1 − odds ratio) × 100.

## Data Availability

Not applicable.

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
