# Peer review of "Effectiveness of Messenger RNA Vaccines against SARS-CoV-2 Infection in Hemodialysis Patients: A Case–Control Study"

_vaccines, 2022, doi:10.3390/vaccines11010049_

Round 1

Reviewer 1 Report

Authors report the clinical findings of using Covid-19 mRNA vaccines in dialysis patients. The study gives an interesting and relevant summary which is highly beneficial to the region (Qatar) while treating dialysis patients and to mitigate the risk of potential Covid-19 infections. The study design is appropriate and the results are presented in a systematic fashion. The conclusions are supported by the findings. The manuscript can be accepted pending few comments as below:

-Introduction part is bit short and authors need to describe the risks of using Covid-19 vaccines against patients with chronic kidney diseases and mechanisms of potential adverse events. This info can be included as part of the second paragraph

-Did the authors measure the levels of antibodies in dialysis patients after they received mRNA vaccines? This would provide important information on the antibody response in dialysis patients which might assist in deciding dosing intervals and frequency. Authors need to discuss this aspect as a limitation of this study

-Also wondering if any dialysis patients were on corticosteroids, which might reduce their immunity further. Please comment on this

-Please check few reference format carefully. eg: Ref 25, first letters in the article title is in lower case. Use consistent format as per the journal

-Page 1 Introduction, line 1 - "was a single mRNA virus" change as "is a single-stranded RNA virus"

Author Response

Thank you for all your valuable comments that helped us improve the quality of our manuscript.

1- Introduction part is bit short and authors need to describe the risks of using Covid-19 vaccines against patients with chronic kidney diseases and mechanisms of potential adverse events. This info can be included as part of the second paragraph

We expanded the introduction added a sentence regarding reported vaccine's side effects.

2- Did the authors measure the levels of antibodies in dialysis patients after they received mRNA vaccines? This would provide important information on the antibody response in dialysis patients which might assist in deciding dosing intervals and frequency. Authors need to discuss this aspect as a limitation of this study

None of our patients had the levels of  antibodies measured post vaccination. There is still no widely accepted standards regarding the desired antibody titers or the assays used to measure SARS-CoV-2 antibodies. We addressed this in detail in the third paragraph of the discussion. We also added it as a limitation of the study.

3- Also wondering if any dialysis patients were on corticosteroids, which might reduce their immunity further. Please comment on this

There were only 2 dialysis patients on corticosteroids and both were cases. This information was added in the results’ section in the revised manuscript. However, no further analysis was done due to the small number of patients.

4- Please check few reference format carefully. eg: Ref 25, first letters in the article title is in lower case. Use consistent format as per the journal

The reference was changed as suggested.

5- Page 1 Introduction, line 1 - "was a single mRNA virus" change as "is a single-stranded RNA virus"

The sentence was changed as suggested.

Reviewer 2 Report

I really thank the editor for the opportunity to review this manuscript. They reported on a large cohort of hemodialysis patients, but I think this paper lack of information related to inclusion criteria and clinical conditions of these patients. However, I have major concerns about the design of the study, as reported below:

Materials & methods:

- Observation period: authors reported an observation period between February 2020 and December 2022; however vaccination policy started later than February 2020. Authors should reported (maybe in the introduction) how was vaccination policy in Qatar and I suggest to start the observation period from the beginning of the vaccination policy.

- “ We used a test-negative case-control design to determine the effectiveness of vaccination against confirmed SARS-CoV-2 infection. This design is a widely accepted standard to assess vaccine effectiveness in a population after introducing a vaccine.”  In this way we cannot exclude that some asymptomatic patients have been included in control groups, and this should be mentioned as a limitation of the study. Moreover to demonstrated that is “a widely accepted standard” some references should be reported by the authors.

Results:

To be honest, I do not think authors’ results introduced new evidence about COIVID 19 infections. We do not know the reason why people were tested for COVID-19 (were they symptomatic?) and therefore we can not be sure of the conclusion. Policy for testing for COVID-19 should be explained and information about severity of the infections should be assessed.

The higher percentage of infection among Asiatics is of interest and should be better addressed. Moreover the reason for kidney failure should be reported and considered s a possible cause of different susceptibility to the infection. To make this paper more interesting, since we are now talking of fouth dose of vaccine, they should not limit the analysis to the  first two doses but they should include data also related to the boosters.

Another point is that we do not know how far from the secon dose they had the infections and this could be relavant for netrpretation of teh results 

Author Response

Thank you for all your valuable comments that helped us improve the quality of our manuscript. 

1- I think this paper lack of information related to inclusion criteria and clinical conditions of these patients. 

“Materials and Methods” section was expanded and more information regarding inclusion and exclusion criteria were added.

2- Observation period: authors reported an observation period between February 2020 and December 2022; however vaccination policy started later than February 2020. Authors should reported (maybe in the introduction) how was vaccination policy in Qatar and I suggest to start the observation period from the beginning of the vaccination policy. 

Vaccination policy in Qatar started in December 2020. We added few sentences regarding vaccination policy under “Materials and Methods” section and we reanalyzed data after changing the observation period starting from December 2020 instead of February 2020. The total number of patients decreased from 1003 to 782 as many unvaccinated patient did not have SARS-CoV-2 test after December 20, 2020 (n=221).

3- We cannot exclude that some asymptomatic patients have been included in control groups, and this should be mentioned as a limitation of the study.

We totally agree that some asymptomatic dialysis patients may have been included in the control groups. We added as a limitation of the study.

4- To demonstrated that is “a widely accepted standard” some references should be reported by the authors. 

We added the following 3 references to the revised manuscript

  1. Chua, H., Feng, S., Lewnard, J.A., Sullivan, S.G., Blyth, C.C., Lipsitch, M., Cowling, B.J. The Use of Test-negative Controls to Monitor Vaccine Effectiveness: A Systematic Review of Methodology. Epidemiology. 2020 Jan;31(1):43-64. doi: 10.1097/EDE.0000000000001116.
  2. Haber, M., Lopman, B.A., Tate, J.E., Shi, M., Parashar, U.D. A comparison of the test-negative and traditional case-control study designs with respect to the bias of estimates of rotavirus vaccine effectiveness. Vaccine. 2018 Aug 9;36(33):5071-5076. doi: 10.1016/j.vaccine.2018.06.072.
  3. Dean, N.E., Hogan, J.W., Schnitzer, M.E. Covid-19 Vaccine Effectiveness and the Test-Negative Design. N Engl J Med. 2021 Oct 7;385(15):1431-1433. doi: 10.1056/NEJMe2113151. Epub 2021 Sep 8. PMID: 34496195; PMCID: PMC8451180.

5- To be honest, I do not think authors’ results introduced new evidence about COIVID 19 infections.

We acknowledge the fact that the effectiveness of COVID-19 vaccine in dialysis population has been reported, but our study provide interesting points that might be of great interest to the readers:

a- We used a new study design (test negative case control) that is recently implemented in this field.

b- We studied variable ethnic background population in a similar environment which is not commonly presented in the literature.

c- High quality data from the Middle East are scarce.

6-  We do not know the reason why people were tested for COVID-19 (were they symptomatic?) and therefore we can not be sure of the conclusion. Policy for testing for COVID-19 should be explained and information about severity of the infections should be assessed.

Most patients were asymptomatic when they got tested. We added few sentences regarding policy of testing under “Materials and Methods” section.    

7- The reason for kidney failure should be reported and considered s a possible cause of different susceptibility to the infection. :

Unfortunately, we didn't not have the etiology of kidney disease in most cases. Most of our patients did not have a kidney biopsy to confirm the etiology of their kidney disease.We instead reported co-morbidities such as diabetes and hypertension in Table 1. Most patients had hypertension (99%) and diabetes mellitus (66%) as reported in table 1.  

8- The higher percentage of infection among Asiatics is of interest and should be better addressed

After reanalyzing data based on the new observation period. Asian race was not found to be statistically significant anymore and this part was removed from “discussion” section.

9- To make this paper more interesting, since we are now talking of fouth dose of vaccine, they should not limit the analysis to the  first two doses but they should include data also related to the boosters.

We expanded our analysis to inclide the third dose of vaccine and we added a paragraph under “Results” section.

10- we do not know how far from the secon dose they had the infections and this could be relavant for interpretation of teh results 

The median time between the second dose and infection was 282 days (IQR: 142-318). This information was added under "Results" section.

 Thank you again and we hope our revised manuscript get accepted for publication.

Reviewer 3 Report

The study submitted by Mohamad M. Alkadi et al., aimed to determine the COVID-19 mRNA vaccine efficacy in SARS-COV-2 patients who are receiving hemodialysis in the State of Qatar. There are several concerns about the study. 

Study population and design is not well written. The first two paragraphs doesn`t fit under the material and method section. As far as I understand, the mRNA vaccine will be available starting from December 23, 2022. How is it possible that authors can collect samples if the mRNA vaccine is not given yet. How authors collected demographic info (age, gender, etc.). Study including criteria, were not given. How do authors decide the vaccine effectiveness in patients? I believe all information including PCR test results were accessed using medical records, but was not clear. Authors need to address this issue upfront in advance, then they need to describe what information was obtained from the medical records including demographics. There is a confusion between abstract and method that did authors include end-stage chronic kidney patients or ambulatory patients maybe both? Besides, what are the defining criteria for chronic HD patients? Authors also determined comorbidities in the results that are mentioned in the method section. Please describe controls in methods too. 

What is the organization name of your local institutional review board? What software was used to obtain medical records (name of the software and version if applicable)? What kind of clinical and lab information was collected from medical records? If the study is done retrospectively and it`s not possible to obtain consent from patients, you might consider obtaining approval from your local committee for a consent waiver and the reason should be addressed by which authors did, but not with a clear statement by the local ethics committee. Is the local ethics committee aware of this? 

Please cite a proper ref for vaccine effectiveness formula. 

Results were poorly written, should be rewritten. 

If authors did not performed the PCR test the results shouldn`t given in results. If it`s required you might need to refer who performed the study or it must be referred to based on medical records. 

Authors did not include patients under age 18 to study but I can see there is an age category given ages between 0–≤40 in the table 1. There is no information available about hepatitis B status in the results, but why include such a statement in introduction and in the discussion? Vaccine effectiveness criteria should be decided based on symptoms rather than the results of infection. Please provide a ref for your decision.  

The discussion can be improved. 

Author Response

Dear Reviewer,

We would like to thank you for all your valuable comments. Please find below point by point response to all your comments:

1- Study population and design is not well written. The first two paragraphs doesn`t fit under the material and method section. Study including criteria, were not given. There is a confusion between abstract and method that did authors include end-stage chronic kidney patients or ambulatory patients maybe both? Besides, what are the defining criteria for chronic HD patients? Authors also determined comorbidities in the results that are mentioned in the method section. Please describe controls in methods too.

Thank you for the valuable comment. We rewrote the “study population and design” section to be more detailed and clear. In this study we only included adult patients with end stage kidney disease patients on chronic hemodialysis defined as being on HD for at least 3 months. Controls are now decribed in the revised methods section.

2- As far as I understand, the mRNA vaccine will be available starting from December 23, 2022. How is it possible that authors can collect samples if the mRNA vaccine is not given yet.

Thank you for your valuable comment and we apologize for the confusion. We wrote the wrong year by mistake and this was corrected to December 23, 2020.

 3- How authors collected demographic info (age, gender, etc.). I believe all information including PCR test results were accessed using medical records, but was not clear. Authors need to address this issue upfront in advance, then they need to describe what information was obtained from the medical records including demographics. What software was used to obtain medical records (name of the software and version if applicable)? What kind of clinical and lab information was collected from medical records?

Thank you for your valuable comment. All data in this study were collected from a national electronic medical record system ((Cerner-North Kansas City, MO, USA)). We added more details in the methods section regarding our electronic medica record system and the different clinical and laboratory parameters that got collected.

 4- What is the organization name of your local institutional review board?

Thank you for your valuable comment. We added in the revised manuscript Hamad Medical Corporation as the name of our local institutional review board.

5- If the study is done retrospectively and it`s not possible to obtain consent from patients, you might consider obtaining approval from your local committee for a consent waiver and the reason should be addressed by which authors did, but not with a clear statement by the local ethics committee. Is the local ethics committee aware of this? 

Thank you for your valuable comment. Our local IRB approved an informed consent waiver and we added a sentence in the revised manuscript regarding the consent waiver.

6- How do authors decide the vaccine effectiveness in patients? Vaccine effectiveness criteria should be decided based on symptoms rather than the results of infection. Please provide a ref for your decision. Please cite a proper ref for vaccine effectiveness formula. 

The test-negative case control design is a widely accepted standard to determine vaccine effectiveness in a population after the introduction of a vaccine. Vaccine effectiveness was determined using the following formula:
Vaccine effectiveness = 1 − Odds(Test+|Vaccinated) / Odds(Test+|Nonvaccinated)
We have used the same design to report overall effectiveness of the Pfizer-BNT162b2 vaccine against the B1.1.7 and B1.351 variants in Qatar and among the US Veterans in the United States.

References:
Abu-Raddad LJ, Chemaitelly H, Butt AA, National Study Group for C-V. Effectiveness of the BNT162b2 Covid-19 Vaccine against the B.1.1.7 and B.1.351 Variants. N Engl J Med. 2021;385(2):187-9.

Butt AA, Omer SB, Yan P, Shaikh OS, Mayr FB. SARS-CoV-2 Vaccine Effectiveness in a High-Risk National Population in a Real-World Setting. Ann Intern Med. 2021;10.7326/m21-1577.

7- Results were poorly written, should be rewritten. 

Thank you for the valuable comment. Results section was rewritten as recommended.

 8- If authors did not performed the PCR test the results shouldn`t given in results. If it`s required you might need to refer who performed the study or it must be referred to based on medical records. 

Thank you for your valuable comment. None of the authors performed PCR tests. PCR tests were done at different healthcare facilities across the whole countries. Authors only reviewed the results of all performed PCR tests in hemodialysis patients using the electronic medical record system.

9- Authors did not include patients under age 18 to study but I can see there is an age category given ages between 0–≤40 in the table 1.

Thank you for your valuable comment. This is true, only adult patients (Age ³ 18) are included in the study. Age variable in table 1 was corrected to 18-≤40.

10- There is no information available about hepatitis B status in the results, but why include such a statement in introduction and in the discussion?

Thank you for your valuable comment. In this study we did not collect data regarding Hepatitis B status, but we mentioned it in the introduction to demonstrate that different vaccines are generally less effective in hemodialysis patients. The statement of Hepatitis B in discussion was removed in the modified manuscript.

 11- The discussion can be improved. 

Thank you for your valuable comments. The discussion was modified as recommended.

Reviewer 4 Report

My main concern with the study is that it gives the impression that COVID-19 mRNA is superior to other type of COVID-19 vaccines even though only one type of vaccine was studied, and no comparisons were made. The authors need to nuance their conclusions accordingly.There is also a repeated mix up of dates: in page 2 under Materials and Methods, paragraph 2 the authors mention that the vaccine  arrived in Qatar on Dec, 23, 2022 . This is impossible since this date has not arrived yet as we write.

Author Response

Dear Reviewer,

We would like to thank you for all your valuable comments. Please find below our response to all the raised concerns:

1- My main concern with the study is that it gives the impression that COVID-19 mRNA is superior to other type of COVID-19 vaccines even though only one type of vaccine was studied, and no comparisons were made. The authors need to nuance their conclusions accordingly.

Thank you for your valuable comment. We added few sentences in the study limitations paragraph to address this point.

 2- There is also a repeated mix up of dates: in page 2 under Materials and Methods, paragraph 2 the authors mention that the vaccine  arrived in Qatar on Dec, 23, 2022 . This is impossible since this date has not arrived yet as we write.

Thank you for your valuable comment and we apologize for the confusion. We wrote the wrong year by mistake and this was corrected in the revised manuscript to December 23, 2020.

Round 2

Reviewer 3 Report

Authors have substantially and satisfactorily revised the MS based on my previous comments. Scientific soundness and quality of the draft have been improved. 

Reviewer 4 Report

The authors have responded satisfactorily to my comments.